# Catalytic asymmetric dearomatization of phenols via divergent intermolecular (3 + 2) and alkylation reactions

Xiang Gao[1], Tian-Jiao Han[1], Bei-Bei Li[1], Xiao-Xiao Hou[1], Yuan-Zhao Hua[1], Shi-Kun Jia[1], Xiao Xiao [2], Min-Can Wang[1], Donghui Wei [1] & Guang-Jian Mei [1] ✉

The catalytic asymmetric dearomatization (CADA) reaction has proved to be a powerful protocol for rapid assembly of valuable three-dimensional cyclic compounds from readily available planar aromatics. In contrast to the well-studied indoles and naphthols, phenols have been considered challenging substrates for intermolecular CADA reactions due to the combination of strong aromaticity and potential regioselectivity issue over the multiple nucleophilic sites (O, C2 as well as C4). Reported herein are the chiral phosphoric acid-catalyzed divergent intermolecular CADA reactions of common phenols with azoalkenes, which deliver the tetrahydroindolone and cyclohexadienone products bearing an all-carbon quaternary stereogenic center in good yields with excellent *ee* values. Notably, simply adjusting the reaction temperature leads to the chemo-divergent intermolecular (3 + 2) and alkylation dearomatization reactions. Moreover, the stereo-divergent synthesis of four possible stereoisomers in a kind has been achieved via changing the sequence of catalyst enantiomers.

The catalytic asymmetric dearomatization (CADA) reaction is a powerful transformation for directly converting planar aromatic compounds into enantioenriched three-dimensional cyclic scaffolds. Over the past decades, significant progress in this field has been made by You and others, which largely contributed to the facile synthesis of natural products and biologically important molecules[1–5]. For example, indole derivatives have been utilized frequently as model substrates for CADA reactions due to their relatively weak aromaticity[6–10]. Naphthols can be treated as conjugated enols in a sense, undergoing CADA transformations to prepare partially dearomatized naphthalenones[11–15]. On the other hand, the CADA reaction of phenols, readily available chemical feedstocks, still remains elusive (Fig. 1A). Oxidative asymmetric dearomatization of substituted phenols can be achieved by using oxidation systems as exampled by the chiral hypervalent iodine/ *m*-CPBA[16–18]. Alternatively, the combination of oxidative dearomatization and subsequent asymmetric desymmetrization has been

employed conventionally as a stepwise strategy for the asymmetric dearomatization of phenols (Fig. 1B)[19–21]. Direct non-oxidative asymmetric dearomatization of phenols is considered challenging, since the regioselectivity issue over the multiple nucleophilic sites (O, C2 as well as C4) is problematic. As such, phenol substrates are often delicately designed by incorporating an electrophilic group to conduct intramolecular dearomative cyclizations (Fig. 1C). For instance, metal-catalyzed intramolecular *para*-position allylation[22,23], arylation[24–27], and alkylation[28] were disclosed by You, Tang and others. Recently, intramolecular *ortho*-position asymmetric dearomatization reactions of phenols have been reported by Ye and others via activation of ynamides or alkynes[29,30]. In a sharp contrast, intermolecular CADA reactions of common phenols, especially the cycloaddition reactions, are far less developed[31].

Given the potential regio-, chemo-, and stereo-selectivity issues associated in the intermolecular CADA reaction of phenols, the

[1]Green Catalysis Center, and College of Chemistry, Zhengzhou University, Zhengzhou 450001, People's Republic of China. [2]Institute of Pharmaceutical Science and Technology, Zhejiang University of Technology, Hangzhou 310014, People's Republic of China. ✉e-mail: meigj@zzu.edu.cn

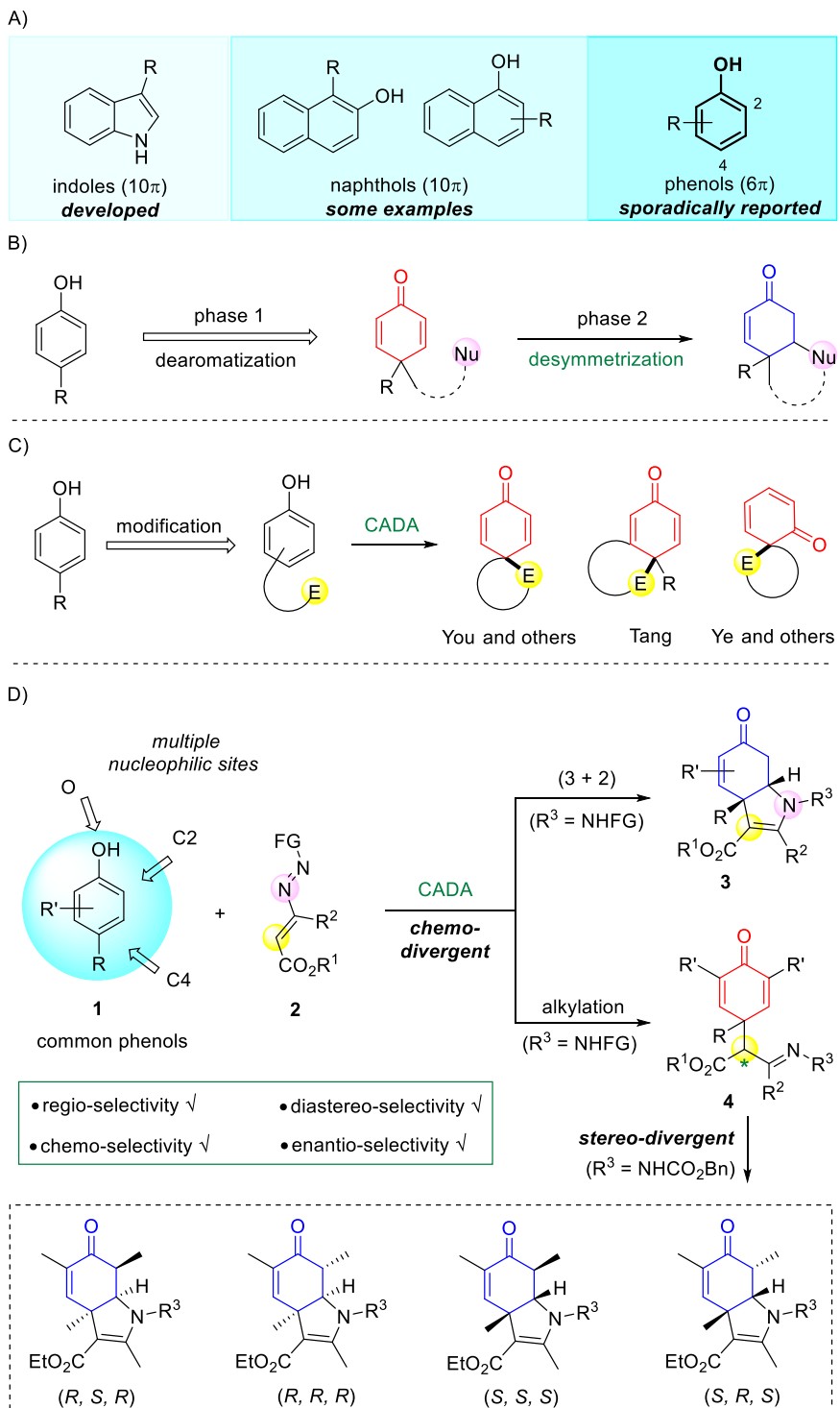

**Fig. 1 | Asymmetric dearomatization of phenols. A** Aromatics in CADA; **B** stepwise strategy; **C** intramolecular strategy; **D** intermolecular (3 + 2) and alkylation reactions (this work).

rational selection of catalytic system and electrophile is key to success. Azoalkenes have emerged as versatile synthons in asymmetric synthesis, as they allow the rapid assembly of valuable N-heterocycles[32,33]. For instance, in situ generated azoalkenes have been employed as four-atom building blocks in copper-catalyzed (4 + n) reactions by Wang and others[34–39]. Recently, we have found that existent azoalkenes could act as three-atom synthons in chiral phosphoric acid (CPA)-catalyzed reacitons[40–43]. Along this line, we report herein the intermolecular CADA reaction of common phenols **1** with azoalkenes **2** under the

catalysis of CPA (Fig. 1D). Notably, this protocol is featured by the temperature-dependent chemo-divergence and the sequence-dependent stereo-divergence[44]. By simply adjusting the reaction temperature, the chemo-divergent intermolecular (3 + 2) and alkylation dearomatization reactions of phenols have been accomplished, which deliver the corresponding products **3** and **4** bearing an all-carbon quaternary stereogenic center in good yields with excellent *ee* values. Meanwhile, with only a single CPA catalyst, changing the use sequence of catalyst enantiomers results in the stereo-divergent

synthesis of four possible stereoisomers in a kind. Additionally, the constructed bicyclic tetrahydroindolone bearing an all-carbon quaternary stereogenic center is the core structure of some bioactive natural alkaloids, such as mesembrine, crinine and powelline[45–49].

## Results

We commenced our investigation with the model reaction between *p*-cresol **1a** and azoalkene **2a** (Table 1). In view of the potential regio- and chemo-selectivity issues, **3a**–**6a** could be the products. Firstly, Brønsted acids were employed. Under the catalysis of **CPA-1** in toluene at 25 °C, intermolecular CADA reaction occurred, delivering the only (3 + 2) product **3a** in a good yield with a moderate enantioselectivity (entry 1). Then, a series of CPAs were screened to find the optimal catalyst. While all the binol-derived CPAs failed to improve the *ee* value (entries 2–6), spinol-derived **CPA-7** gave an encouraging result wherein the *ee* value was increased to a relatively higher level (entry 7). So, **CPA-7** was selected as the best catalyst for the subsequent condition screening. Solvent effect was studied, indicating that toluene was the best choice (entry 7 vs 8–10). Lowering reaction temperature contributed to the enantio-control (entries 11–12). Finally, additives were used to further optimize the reaction. With 5 Å molecular sieve, both the yield and *ee* value of product **3a** could be improved to a high level (entry 14). Interestingly, Lewis acids led to a different regioselective result (see Supplementary Information for details). For example, while AgOTf offered the (3 + 2) dearomatization product **3a** (entry 16), AgOAc resulted in the *oxa*-Michael addition product **5a** (entry 15). We tried to control enantioselectivity of this *oxa*-Michael process by using silver chiral phosphate. However, only dearomative (3 + 2) product **3a** was obtained (entry 17), which indicated silver chiral phosphate could also catalyze the dearomatization process. Notably, none of the C4-dearomatization alkylation product **4a** and C2-Friedel-Crafts alkylation product **6a** were observed during the optimization.

With the best reaction conditions in hand, the reaction generality of azoalkenes was examined. As shown in Fig. 2, the reaction was applicable to a variety of azoalkenes. The *N*-protective group could be varied from -CO₂Bn (**3a**) to -CO₂Me (**3b**), -CO₂Et (**3c**), and -CO₂fluorenylmethyl (**3d**) with consistently excellent *ee* values. The ester moiety appended to the C=C double bond had little effect on the reaction efficiency, all the tested substrates (**3e**–**3j**) were well-tolerated. In contrast, the R² group significantly affected the enantioselectivity of the reaction. The larger the R² group, the lower the *ee* values (**3k**–**3l**).Next, we turned our attention to the substrate scope of phenols (Fig. 2). Various phenols bearing alkyl substituents at 4-position, such as ethyl (**3m**), *n*propyl (**3n**), *n*butyl (**3o**) and *n*pentyl (**3p**) groups as well as allyl group (**3q**), served as suitable substrates for this CADA (3 + 2) reaction, allowing the installation of different moieties at the quaternary stereogenic center. In addition, the hindered cyclopropyl group (**3r**) worked equally well under standard conditions. Notably, the aryl group at the 4-position was also tolerated, as product **3s** was readily prepared. Furthermore, we examined the tolerance of the reaction to di-substituted phenols. Products **3t**–**3z** were prepared in good yields with good to excellent enantioselectivities under the modified conditions. Among them, the alkyl groups at the C2-position were in favor of enantio-control (**3t**–**3v**), while methoxy group resulted in a slight decrease in *ee* value (**3w**). Substituents at the C3-position had some delicate influence on the reaction. For example, **3x** and **3y** were obtained in good *ee* values, while **3z** was formed in an excellent enantioselectivity. A regio-isomer was observed in the preparation of **3x** (see Supplementary Information for details). Remarkably, the projected reaction was applicable to 1-naphthol, leading to the corresponding product **3a′** in good yield and *ee* value. The absolute configurations of the products were assigned based on X-ray crystallographic analysis of **3t**.

Encouraged by the success of CADA (3 + 2) reaction of mono- and di-substituted phenols, we then studied the substrate scop by

employing tri-substituted phenols. Unexpectedly, the temperature-dependent chemo-divergent dearomatization reactions of phenol **1o** were observed (Fig. 3A). At −30 °C, the standard conditions delivered the only dearomative alkylation product **4b** in an excellent yield and with an excellent *ee* value. On the other hand, the CADA (3 + 2) reaction product **8** was obtained at 25 °C. Given the synthetic and biological importance of chiral cyclohexadienones, the substrate scope of this intermolecular catalytic asymmetric alkylation dearomatization had been studied (Fig. 3B). The examination of various azoalkenes indicated that the R² group had some influence on the reaction efficiency. Reactions using methyl substituted azoalkenes (R² = Me) were conducted smoothly, affording the corresponding cyclohexadienone products **4c**–**4i** in good yields with excellent enantioselectivities (76–93% yields, 96–>99% *ee*). However, the steric hindered ethyl (**4j**), *n*butyl (**4k**), and benzyl (**4l**) groups were not well compatible, which was same as the (3 + 2) process. The generality of tri-substituted phenols was also investigated. While electron-donating methoxy group (**4m**) afforded the consistently good yield and *ee* value, electron-withdrawing bromo group (**4n**) gave the result of no reaction.

To gain insights into the reaction mechanism, some control experiments were performed (Fig. 4). Firstly, enantioenriched **4b** was subjected to the standard conditions at room temperature. Product **8** was formed in an excellent yield and with good stereoselectivities (Fig. 4A), which clearly suggested that **8** is a productive intermediate for the (3 + 2) reaction. More importantly, with the same enantioenriched **4b**, (*S*)-**CPA-7** led to a different diastereoisomer **9** (Fig. 4B), which implied the substrate control for the enantioselectivity and catalyst control for the diastereoselectivity. These results paved the way to a sequence-dependent stereo-divergent synthesis enabled by one chiral catalyst. Namely, changing the use sequence of (*R*)-**CPA-7** and (*S*)-**CPA-7** could result in the stereo-divergent synthesis of four possible stereoisomers in a kind. To demonstrate this concept, four possible stereoisomers of the (3 + 2) reaction products were prepared from **1o** and **2a** (Fig. 4E). However, under the catalysis of (±)-**CPA-7**, racemization of **4b** was observed which led to a low *ee* value of the cyclizaiton product (Fig. 4C). On the other hand, when racemic **4b** was employed, the enantio-convergent desymmetrization of cyclohexadienone was achieved (Fig. 4D), which will be fully investigated in the future. Based on these experimental results, a plausible reaction pathway and possible transition states (TS) were suggested in Fig. 4F. In TS-1, **CPA-7** simultaneously activated phenol **1a** and azoalkene **2a** via dual hydrogen-bonding interactions. The CPA-catalyzed asymmetric dearomative alkylation of phenol **1a** with azoalkene **2a** occurred, generating the hydrazone intermediate **Int-1** with a chiral carbon center. Subsequently, the group selectivity was accomplished via substrate-controlled enantioselective intramolecular cyclization. While pathway *a* led to the unfavored iminium **Int-2** due to the sterically hindered *cis*-orientation of methyl and ester groups, pathway *b* gave the favored one **Int-3**. The final CPA-catalyzed diastereoselective proton transfer via tautomerization facilitated the (3 + 2) process and delivered the dearomative product **3a**.

Finally, to examine the practicality of this protocol, a convenient gram-scale synthesis of **3u** was conducted under standard reaction conditions (see Supplementary Information for details). Then a series of derivatizations were performed, further highlighting the synthetic utility (Fig. 5). Firstly, the enone motif could be manipulated via 1,2-addition, 1,4-hydrogenation and 1,2-reduction affording the corresponding product **10**–**12** in good yields and without erosion of enantiomeric purity. Besides, the protecting group on the azo nitrogen center could be easily removed, yielding product **13** in a good yield of 93%. Notably, *N*-allylic alkylation of **3a** led to the formation of compound **14** incorporating a N−N axially stereogenic axis. The N−N bond cleavage had been achieved via a base-promoted eliminative strategy, furnishing the amine product **15**.

**Table 1 | Reaction optimization[a]**

| Entry | Catalyst | Solvent | T (°C) | Product | Yield (%)[b] | ee (%)[c] |
|---|---|---|---|---|---|---|
| 1 | CPA-1 | Toluene | 25 | 3a | 62 | 67 |
| 2 | CPA-2 | Toluene | 25 | 3a | 78 | 34 |
| 3 | CPA-3 | Toluene | 25 | 3a | 70 | 40 |
| 4 | CPA-4 | Toluene | 25 | 3a | n.r. | – |
| 5[d] | CPA-5 | Toluene | 25 | 3a | 87 | 61 |
| 6 | CPA-6 | Toluene | 25 | 3a | 43 | 29 |
| 7 | CPA-7 | Toluene | 25 | 3a | 81 | 80 |
| 8 | CPA-7 | CH$_2$Cl$_2$ | 25 | 3a | 66 | 19 |
| 9 | CPA-7 | CH$_3$CN | 25 | 3a | n.r. | – |
| 10 | CPA-7 | THF | 25 | 3a | n.r. | – |
| 11 | CPA-7 | Toluene | -20 | 3a | 79 | 82 |
| 12 | CPA-7 | Toluene | -30 | 3a | 80 | 89 |
| 13[e] | CPA-7 | Toluene | -30 | 3a | 78 | 92 |
| 14[f] | CPA-7 | Toluene | -30 | 3a | 80 | 92 |
| 15 | AgOAc | Toluene | 25 | 5a | 73 | – |
| 16 | AgOTf | Toluene | 25 | 3a | 80 | – |
| 17 | Ag$_2$CO$_3$/CPA-7 | Toluene | 25 | 3a | 75 | 74 |

ee enantiomeric excess, n.r. no reaction, THF tetrahydrofuran.

[a]Unless indicated otherwise, 1a (0.05 mmol), 2a (0.06 mmol) and catalyst (10 mol%) in a solvent (1 mL) at the specified temperature, all dr >20:1.

[b]Isolated yields.

[c]Determined by chiral HPLC analysis.

[d]Catalyst with opposite configuration was used.

[e]50 mg 3 Å molecular sieve was used.

[f]50 mg 5 Å molecular sieve was used.

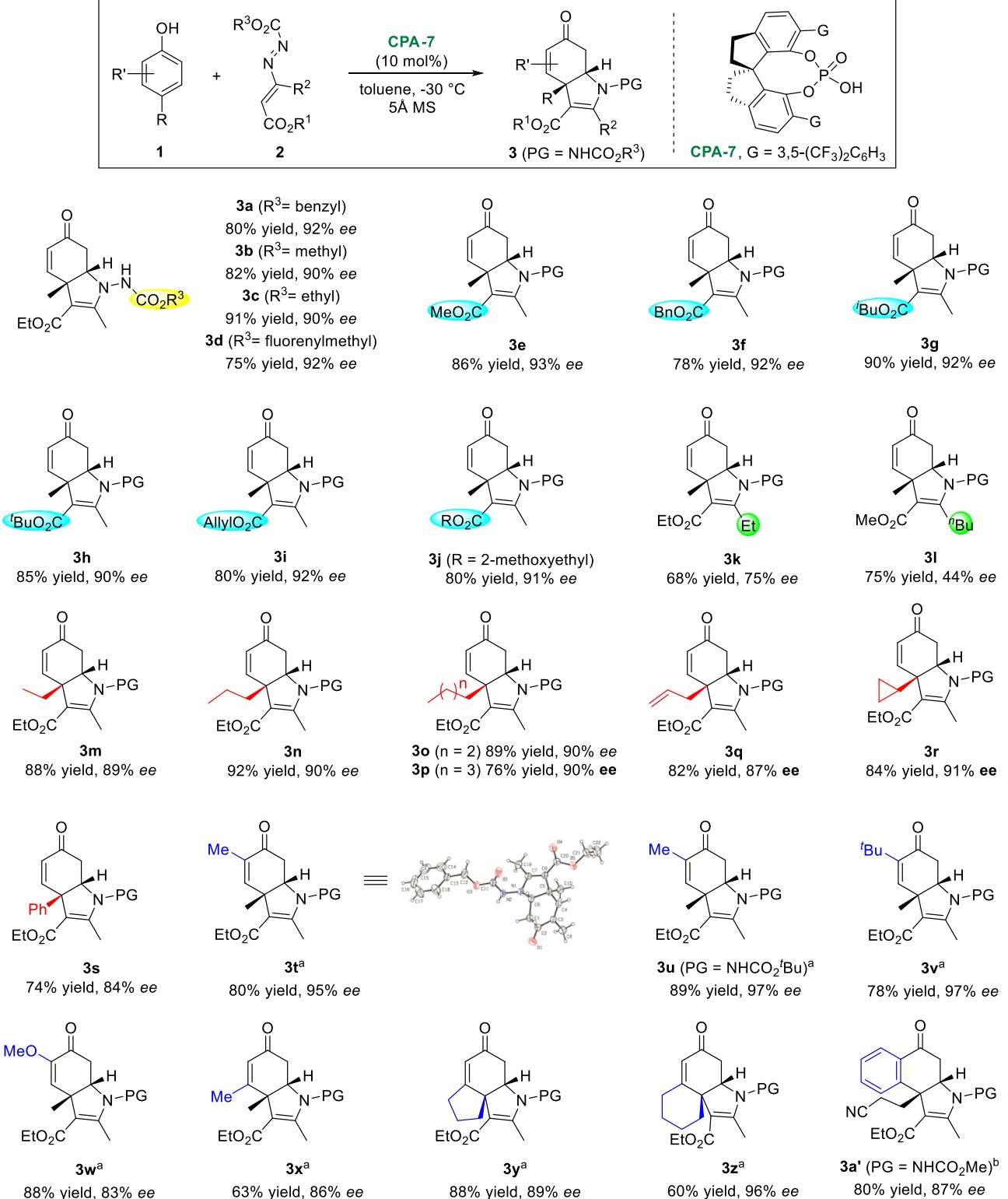

**Fig. 2 | Substrate scope for CADA (3 + 2) reaction.** Reaction conditions: **1** (0.2 mmol), **2** (0.24 mmol.), **CPA-7** (10 mol%), and 5 Å MS (50 mg) in toluene (1 mL) at −30 °C for 12 h, isolated yields, *ee* and dr values were determined by chiral HPLC analysis, all dr >20:1, PG = NHCO₂Bn. [a]with 3 Å MS (50 mg) and **CPA-2** at −40 °C. [b]with 3 Å MS (50 mg) and **CPA-4** at 25 °C.

## Discussion

In conclusion, we have established the CPA-catalyzed divergent intermolecular CADA reactions of common phenols with azoalkenes, which deliver the tetrahydroindolone and cyclohexadienone products bearing an all-carbon quaternary stereogenic center in good yields with excellent *ee* values. As opposite to the previous dearomatization reactions involving indoles and naphthols, this work employs readily available phenols, which have been considered challenging substrates for intermolecular asymmetric dearomatization reactions due to the combination of strong aromaticity and the potential regioselectivity

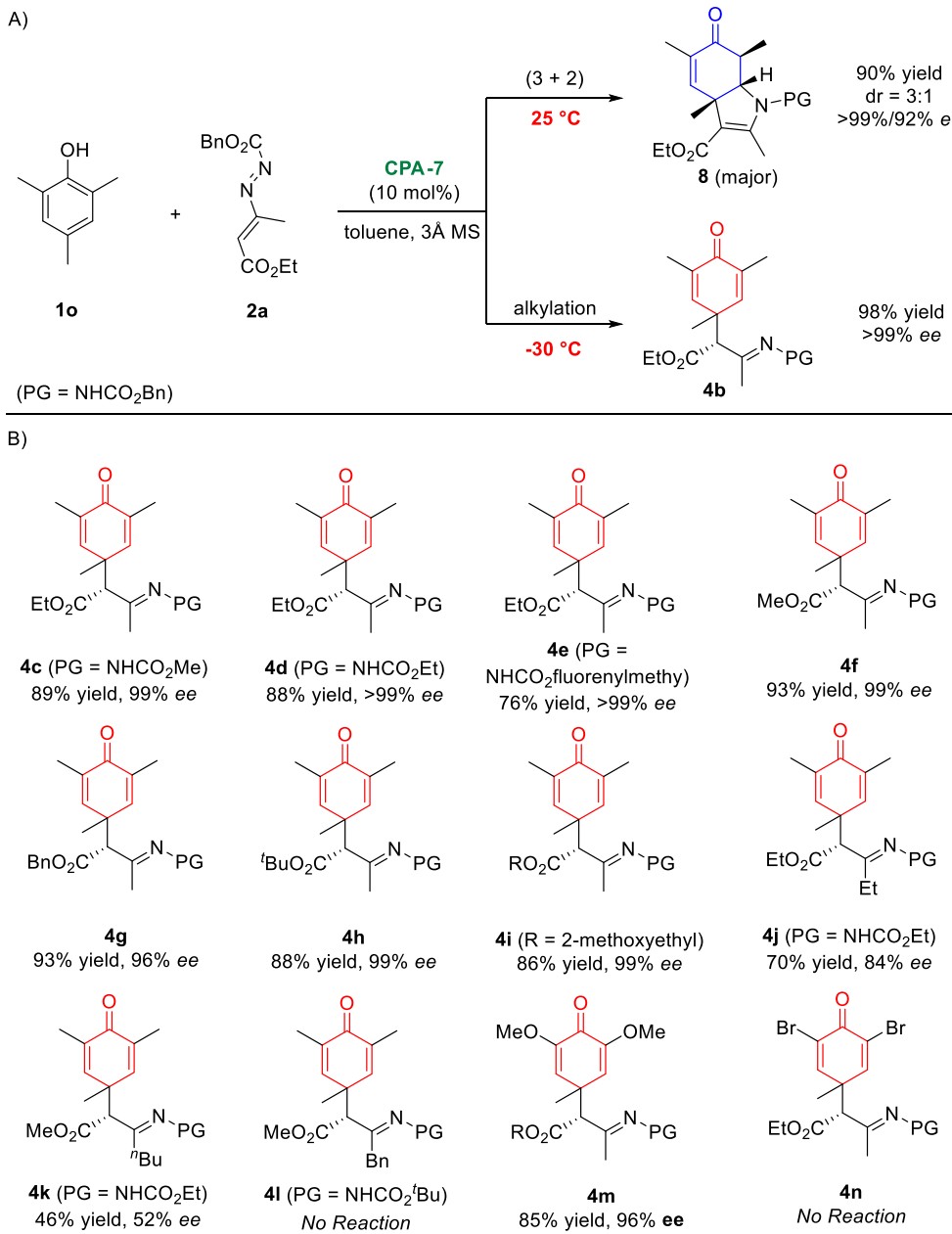

**Fig. 3 | Chemo-divergent (3 + 2) and alkylation dearomatization reactions. A** Reaction discovery; **B** substrate scope. Reaction conditions: **1** (0.2 mmol), **2** (0.24 mmol.), **CPA-7** (10 mol%), and 3 Å MS (50 mg) in toluene (1 mL) at −30 °C for 12 h, isolated yields, *ee* value was determined by chiral HPLC analysis.

issue over the multiple nucleophilic sites (O, C2 as well as C4). Notably, this reaction is featured by the temperature-dependent chemo-divergence and the sequence-dependent stereo-divergence. Simply adjusting the reaction temperature leads to the chemo-divergent intermolecular (3 + 2) and alkylation dearomatization reactions. More importantly, stereo-divergent synthesis of four possible stereoisomers in a kind has been achieved by changing the sequence of catalyst enantiomers. Further applications of this protocol are currently ongoing in our laboratory and will be reported in due course.

## Methods
### Typical procedure for (3 + 2) reaction
Para substituted phenols **1** (0.20 mmol), 5 Å MS (50 mg), **(*R*)-CPA-7** (10 mol%) were dissolved in toluene (1 mL), and azoalkenes **2** (0.24 mmol) were added dropwise at −30 °C. The reaction mixture was stirred at the same temperature for 12 h. The solvent was removed in vacuo and the crude product was separated by flash column

chromatography on silica gel (petroleum ether/ethyl acetate 3:1–1:1) to afford the products **3**.

### Typical procedure for alkylation reaction
Tri-substituted phenols **1** (0.20 mmol), 3 Å MS (50 mg) and **(*R*)-CPA-7** (10 mol%) were dissolved in toluene (1 mL), and azoalkenes **2** (0.24 mmol) were added dropwise at −30 °C. The reaction mixture was stirred for 12 h. The solvent was removed in vacuo and the crude product was separated by flash column chromatography on silica gel (petroleum ether/ethyl acetate 3:1–1:1) to afford the products **4**.

### Data availability
The authors declare that the data relating to the characterization of products, experimental protocols and the computational studies are available within the article and its Supplementary Information file. The X-ray crystallographic coordinates for structures reported in this study have been deposited at the Cambridge Crystallographic Data Centre

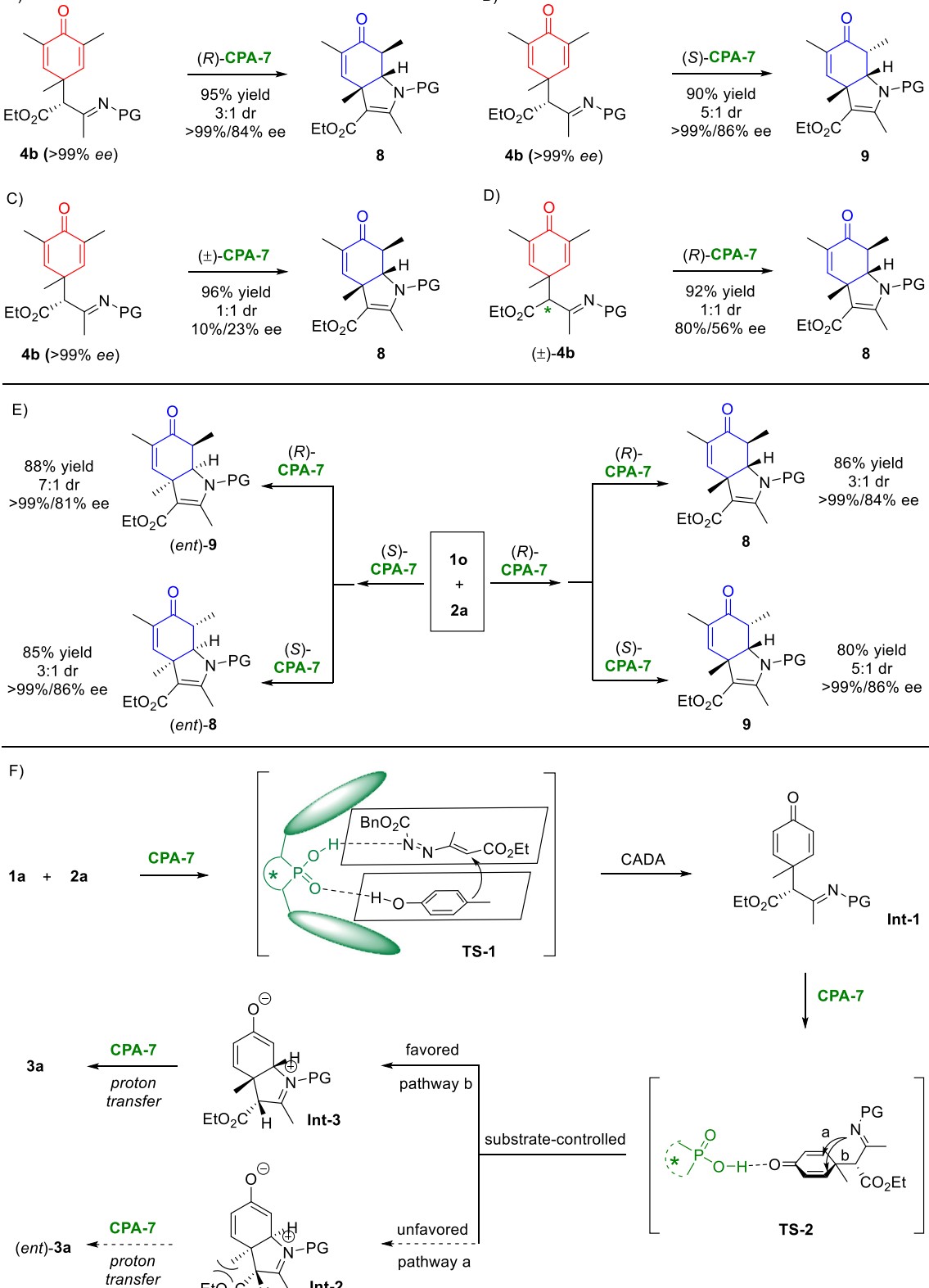

**Fig. 4 | Mechanistic considerations and stereo-divergent synthesis.**
**A** Cyclization of enantioenriched **4b** with (*R*)-**CPA-7**; **B** cyclization of enantioen-riched **4b** with (*S*)-**CPA-7**; **C** cyclization of enantioenriched **4b** with racemic **CPA-7**; **D** Cyclization of racemic **4b** with (*R*)-**CPA-7**; **E** stereo-divergent synthesis; **F** proposed reaction pathway and transition states.

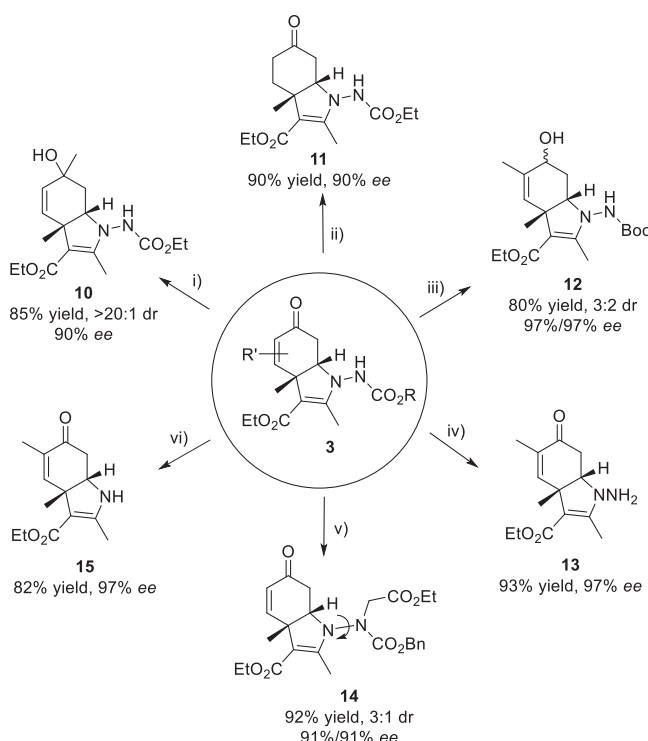

**Fig. 5 | Preliminary applications.** Reaction conditions: i) **3c** (0.2 mmol), CH$_3$MgBr (0.3 mmol.), THF, 0 °C; ii) **3c** (0.3 mmol), Pd/C, H$_2$, MeOH, r.t.; iii) **3u** (0.2 mmol), NaBH$_4$ (0.2 mmol), MeOH, 0 °C; iv) **3 u** (0.2 mmol), TFA (0.2 mL), CH$_2$Cl$_2$, 0 °C; v) **3a** (0.2 mmol), ethyl 2-bromoacetate (0.24 mmol), Cs$_2$CO$_3$ (0.3 mmol), CH$_3$CN, r.t.; vi) **3u** (0.2 mmol), ethyl 2-bromoacetate (0.24 mmol), Cs$_2$CO$_3$ (0.3 mmol), CH$_3$CN, r.t. then reflux.

(CCDC), under deposition numbers CCDC 2216396 for (−)-**3t** and CCDC 2203087 for (±)-**3t**. These data can be obtained free of charge from The Cambridge Crystallographic Data Centre via www.ccdc.cam. ac.uk/data_request/cif.

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

## Acknowledgements

We are grateful for the generous financial support from Natural Science Foundation of Henan Province (222300420084 to G.-J.M.), application research plan of Key Scientific Research Projects in Colleges and Universities of Henan Province (22A150056 to G.-J.M.), and National Natural Science Foundation of China (22208302 to X.X.).

## Author contributions

X.G. and T.-J.H. performed and analyzed the experiments. B.-B.L., Y.-Z.H., S.-K.J., X.X. and M.-C.W. participated in the early development of the project. X.-X.H. and D.W. did the calculations. G.-J.M. conceived and designed the project. G.-J.M. overall supervised the project. All authors prepared this manuscript.

## Competing interests

The authors declare no competing interests.
