## [Peer Review File · Nature Communications]

Catalytic Asymmetric Dearomatization of Phenols via Divergent Intermolecular (3 + 2) and Alkylation ReactionsReviewers' Comments:

Reviewer #1:

Remarks to the Author:

In this manuscript, Mei and co-workers describe an enantioselective synthesis of bicyclic tetrahydroindolones with chiral phosphoric acid as the bifunctional catalyst by intermolecular [3+2] cyclization reaction of phenols with azoalkenes. Under rather mild conditions, the reaction generally proceeds in excellent yields and enantioselectivities. Of particular note, the substrate scope is rather general for both substituted phenols with azoalkenes. The cyclization reaction is a step-wise process, which has been proved by the isolation of alkylation products 4 in excellent yields and ees by lowering the reaction temperature. In addition, the stereodivergent synthesis by using different enantiomers (Scheme 4) in different stage is highly interesting.

This is a truly interesting CADA reaction as the previous reports focused on indoles and naphthols.

This reviewer believes the current results are highly interesting and inspiring to the field of asymmetric catalysis in general. Therefore publication in Nature Commun is highly recommended.

The following minor comments should be addressed prior to publication:

(1) In the abstract, second line, "planner" to "planar"

(2) A closely related CADA reaction of naphthol enabled by CPA at the 4-position should be cited: Xia, Z.-L.; Zheng, C.; Xu, R.-Q.; You, S.-L. Chiral Phosphoric Acid Catalyzed Aminative Dearomatization of α -Naphthols/Michael Addition Sequence. Nat. Commun. 2019, 10, 3150.

Reviewer #2:

Remarks to the Author:

The manuscript by Mei and coworkers describes a case of a robust system for chiral phosphoric acid-catalyzed divergent intermolecular CADA reaction of phenols with azoalkenes that provides a highly efficient synthesis of tetrahydroindolones and cyclohexadienones bearing an all-carbon quaternary stereogenic center in good yields and enantioselectivity. Notably, the stereo-divergent synthesis of four possible stereoisomers has been achieved via changing the sequence of catalyst enantiomers. A series of derivatizations were performed, further highlighting the synthetic utility.

In my view, this finding is significant enough to be published in the Nature Communications; however, there are several issues with the manuscript that the authors should address prior to the publication. These issues are listed below.

1) The "first" can be found in the manuscript, any publishable results can find their "first". It really doesn't make any sense to spotlight "first". As such, "first" should also be removed from the manuscript.

2) The high selectivity achieved for the product 3z is remarkable. To demonstrate the universality of this method, a few examples of 1-naphthol substrates should be added.

3) Various phenols bearing alkyl substituents at 4-position were successfully achieved, how about phenols containing aryl substituents at 4-position?

Reviewer #3:

Remarks to the Author:

Mei and coworkers have presented a fascinating study on the catalytic asymmetric dearomatization of para-substituted phenols using chiral phosphoric acid catalysis. The majority of the target products were obtained with high chemo- and stereo-selectivity, and the control experiments to some extent provided insights into the reaction mechanism. This method represents a significant advance in the field of organocatalytic asymmetric dearomatization. In my opinion, after addressing the following issues and providing more attractive results, this manuscript can be published in Nature Communications.

1. Minor errors need to be thoroughly checked and corrected to ensure the manuscript's accuracy.

2. The experimental data only involved phenols substituted with alkyl groups in the para position. It would be valuable to explore other substituents besides those that donate electrons.
3. The high enantioselectivity in the CADA (3+2) reaction was only observed for R₂ = Me. It is recommended to test other substituents, not just alkyl ones, for the alkylation dearomatization reactions in Scheme 3.
4. In Scheme 4, the mechanism for enantioselective intramolecular cyclization needs a more detailed and accurate description.
5. References for the application of bicyclic tetrahydroindolone into Ref. 43-45 need to be added. Closely related research in *Angew. Chem. Int. Ed.* 2017, 56, 3242 needs to be cited. Moreover, the preliminary applications presented in Scheme 5 are not particularly compelling. If some interesting applications can be added, it will further enhance the persuasiveness of this method.

Point-to-Point Responses to Reviewers' Comments (Manuscript ID: NCOMMS-23-13058-T)

Please take note that all the descriptive, positive comments of the reviewers are omitted, and only the reviewers' comments expressing their concerns/suggestions are listed below, which are followed by our responses. All the changes made in the revised manuscript as highlighted in yellow.

Revisions made in reply to Reviewer 1's comments:

- **Comment #1:** "In the abstract, second line, "planner" to "planar"."
Our response: It has been revised accordingly.
- **Comment #2:** "A closely related CADA reaction of naphthol enabled by CPA at the 4-position should be cited: Xia, Z.-L.; Zheng, C.; Xu, R.-Q.; You, S.-L. Chiral Phosphoric Acid Catalyzed Aminative Dearomatization of α -Naphthols/Michael Addition Sequence. Nat. Commun. 2019, 10, 3150."
Our response: The above-mentioned ref. has been properly cited in ref. 15, see the revised manuscript.

Revisions made in reply to Reviewer 2's comments:

- **Comment #1:** "The "first" can be found in the manuscript, any publishable results can find their "first". It really doesn't make any sense to spotlight "first". As such, "first" should also be removed from the manuscript."
Our response: The word "first" has been removed accordingly.
- **Comment #2:** "The high selectivity achieved for the product 3z is remarkable. To demonstrate the universality of this method, a few examples of 1-naphthol substrates should be added."
Our response: Thank you for this good suggestion. 4-Methyl 1-naphthol has been tested accordingly. However, it failed to afford high *ee* values. We have added this result into SI, since the main text focuses on the dearomatization of phenols.
- **Comment #3:** "Various phenols bearing alkyl substituents at 4-position were successfully achieved, how about phenols containing aryl substituents at 4-position?"

Our response: Thank you for this good comment. 4-Phenyl phenol has been employed accordingly. The reaction readily occurred under the standard conditions, giving the corresponding product in 74% yield and 84% *ee*, see the updated Scheme 2.

Revisions made in reply to Reviewer 3's comments:

- **Comment #1:** “Minor errors need to be thoroughly checked and corrected to ensure the manuscript's accuracy.”

Our response: The manuscript has been carefully re-checked, and mistakes have been revised accordingly.

- **Comment #2:** “The experimental data only involved phenols substituted with alkyl groups in the para position. It would be valuable to explore other substituents besides those that donate electrons.”

Our response: It's a similar comment with the Comment #3 of reviewer 2. 4-Phenyl phenol has been employed accordingly. The reaction readily occurred under the standard conditions, giving the corresponding product in 74% yield and 84% *ee*, see the updated Scheme 2. In fact, this reaction is sensitive to the electronic property of phenol substrate. Phenols with electron-withdrawing groups are not applicable. For example, no desired product was obtained by utilizing 4-floro, 4-chloro, and 4-ester groups, and the decomposition of azoalkene was observed. We have added these results into the revised SI.

- **Comment #3:** “The high enantioselectivity in the CADA (3+2) reaction was only observed for R² = Me. It is recommended to test other substituents, not just alkyl ones, for the alkylation dearomatization reactions in Scheme 3.”

Our response: Thank you for this good suggestion. Other substituents (R²) have been tested accordingly, see the revised Scheme 3. The examination of various azoalkenes indicated that the R² group had some influence on the reaction efficiency. Reactions using methyl substituted azoalkenes (R² = Me) were conducted smoothly, affording the corresponding cyclohexadienone products **4c–4i** in good yields with excellent enantioselectivities (76–93% yields, 96–>99% *ee*). However, the steric hindered ethyl (**4j**), ⁿbutyl (**4k**), and benzyl (**4l**) groups were not well compatible, which was same as the (3 + 2) process. Azoalkenes are highly active, only alkyl substituted (R²) azoalkenes are stable enough to be prepared. So far, we have not successfully synthesized azoalkenes with non-alkyl groups which are almost not reported in literatures. This is a limitation for azoalkene-involved reactions, and we will focus on addressing this issue.

- **Comment #4:** “In Scheme 4, the mechanism for enantioselective intramolecular cyclization needs a more detailed and accurate description.”

Our response: More detailed description for the intramolecular cyclization step has been added, see the revised Scheme 4 and corresponding text.

- **Comment #5:** “References for the application of bicyclic tetrahydroindolone into Ref. 43-45 need to be added. Closely related research in *Angew. Chem. Int. Ed.* 2017, 56, 3242 needs to be cited. Moreover, the preliminary applications presented in Scheme 5 are not particularly compelling. If some interesting applications can be added, it will further enhance the persuasiveness of this method.”

Our response: Thank you for this comment. Two references on the application of bicyclic tetrahydroindolone in natural products synthesis have been properly cited in ref. 48 and 49. The above-mentioned reference (*Angew. Chem. Int. Ed.* 2017, 56, 3242) have been properly cited in ref. 14. The intermolecular CADA reaction of common phenol is a highly challenging task in organic synthesis. In this manuscript, we focus on the methodology development, including chemo-divergent (3 + 2) and alkylation, stereo-divergent synthesis of four possible stereoisomers, and also some preliminary applications. Currently, we are trying to synthesize some complex molecules including natural alkaloids and pharmaceuticals using this protocol. However, these tasks are still ongoing and require more time. Therefore, these progresses will be reported independently.

Reviewers' Comments:

Reviewer #2:

Remarks to the Author:

I am satisfied with the revisions. This paper can be published in NC.

[Note from the Editor: Reviewer #2 was asked to look also over the response given to reviewer #3 and thinks that the authors carefully answered the reviewer's questions]